# Using the Spark Plasma Sintering System for Fabrication of Advanced Semiconductor Materials

**DOI:** 10.3390/ma17061422

**Published:** 2024-03-20

**Authors:** Kamil Kaszyca, Marcin Chmielewski, Bartosz Bucholc, Piotr Błyskun, Fatima Nisar, Jerzy Rojek, Rafał Zybała

**Affiliations:** 1Lukasiewicz Research Network, Institute of Microelectronics and Photonics, al. Lotnikow 32/46, 02-668 Warsaw, Poland; marcin.chmielewski@imif.lukasiewicz.gov.pl (M.C.); bartosz.bucholc@imif.lukasiewicz.gov.pl (B.B.); 2Institute of Fundamental Technological Research, Polish Academy of Sciences, Pawinskiego 5B, 02-106 Warsaw, Poland; fnisar@ippt.pan.pl (F.N.); jrojek@ippt.gov.pl (J.R.); 3Faculty of Materials Science and Engineering, Warsaw University of Technology, Woloska 141, 02-507 Warsaw, Poland; piotr.blyskun@pw.edu.pl

**Keywords:** spark plasma sintering, arc melting, semiconductor materials, half-Heusler, bismuth telluride, cobalt triantimonide, SHS, SPS

## Abstract

The interest in the Spark Plasma Sintering (SPS) technique has continuously increased over the last few years. This article shows the possibility of the development of an SPS device used for material processing and synthesis in both scientific and industrial applications and aims to present manufacturing methods and the versatility of an SPS device, presenting examples of processing Arc-Melted- (half-Heusler, cobalt triantimonide) and Self-propagating High-temperature Synthesis (SHS)-synthesized semiconductor (bismuth telluride) materials. The SPS system functionality development is presented, the purpose of which was to broaden the knowledge of the nature of SPS processes. This approach enabled the precise design of material sintering processes and also contributed to increasing the repeatability and accuracy of sintering conditions.

## 1. Introduction

Currently, materials engineers are mostly focused on the development of new materials that are characterized by higher operational parameters. However, this also forces the development of materials fabrication methods that overcome the limitations of conventional fabrication methods. This work presents the versatility (both in application and upgrade potential) of an SPS device using as examples three main topics listed below:
An example of compacting grounded arc-melted materials with an SPS device (research performed before SPS upgrade).An original approach to an SPS system extension, allowing a detailed measurement of the sintering system.An approach to use an SPS device as an SHS reactor, which is possible due to multi-point parameter measurement.

The SPS devices are mostly used as compacting devices, allowing one to fabricate bulk, solid materials from powders using sintering phenomena—in most cases the modification of phase composition is undesirable. One of the topics presented in this work shows the use of SPS as a reactor, where synthesis actually occurs. During the synthesis process, the phase composition of the material is intentionally changed.

The increasing popularity of the SPS method [1] and its increased accessibility is visible in many publications that use SPS as the main method for material forming and densification. The ability to rapidly heat and cool opens the opportunities of novel materials preparation. The SPS technique allows one to obtain a compact from powder in just one production cycle. This enables the synthesis and formation of elements from materials sensitive to the environment, e.g., advanced semiconductor materials. Due to short sintering times, materials fabricated via the spark plasma sintering processes exhibit much less grain growth and a lower probability of decomposition.

The diagram of the SPS equipment is shown in Figure 1. Processed material (3) is placed in a graphite die (1) and closed by two graphite punches (2). Then, the die and the powder are placed between the device’s electrodes in the vacuum chamber (5).

The current flowing through the die and powder causes the release of a large amount of Joules of heat, heating the entire system. In some cases, during SPS, arcing may occur between the sintered particles.

Figure 2 presents the diagram of the SPS parameters taking part in the sintering process. The model presented in work [2] takes into account the geometry of the sintering elements and mentions basic process parameters, but simplifies the issue to a simple temperature/voltage/displacement functions output. The more advanced approach is presented in work [3], where the authors compare grain growth as a function of temperature in transparent ceramics. Work [4] presents the comparison of simulated and experimental data of SPS samples and applies the temperature-gradient issue to the created model. The cited works show the increase in complexity of research connected with modeling and justify the necessity of the SPS system functionality expansion. All of the presented works focuses on modeling—the real collected data are rarely presented and discussed.

The basic, minimal set of information (process parameters) describing the SPS densification process is the following:
Annealing temperature—the highest maintained temperature over a process.Pressure—uniaxial pressure applied to the sample during the sintering process.Dwell time—sintering temperature holding time.

The presented parameters are sufficient in most cases [5,6,7], but the development of mathematical models of the sintering process, novel techniques and materials need to be supported by a detailed description of sintering process parameters over time. This article shows the possibility of the SPS device development and its application for semiconductor materials processing or synthesis in both scientific and industrial applications. The versatility and captivity of the SPS technique make it a preferred solution for many different materials including biomaterials, nanomaterials [8] and semiconductors.

The thermoelectric phenomenon allows for direct conversion of heat into electrical energy. It occurs in almost all materials, but only a small group of semiconductors exhibit properties allowing for their practical usage for energy generation. Their conversion usefulness is parameterized by a figure-of-merit factor [9] (ZT) as defined in Equation (1).
(1)$$ZT=\frac{\alpha^{2}\times \sigma }{\kappa_{e}+\kappa_{l}}$$
A good thermoelectric material exhibits a high Seebeck coefficient (α), high electrical conductivity (σ) and low heat conductivity (κ). Such a combination of properties is rare and advanced fabrication methods have to be used to achieve it. The heat conductivity κ consists of two components: electron part $\kappa_{e}$ and lattice part $\kappa_{l}$. There are two widely used mechanisms of ZT factor optimization: (1) doping the material to improve the Seebeck coefficient and electrical conductivity [10], and (2) material structurization, which has the highest impact on lattice thermal $\kappa_{l}$ conductivity [11]. The reduction in heat conductivity can also occur if the dopant atoms significantly differ from the main semiconductor structure (e.g., there is a significant difference in atomic mass). This leads to a phenomenon called phonon scattering [12,13] and it is a promising effect, causing heat conductivity reduction.

The first part of this work aims to show the possibilities of the SPS device extension and present example materials we have processed using our device. The most common methods of thermoelectric materials synthesis are direct synthesis of melted substrates [6,14,15] and mechanical alloying [14,16]. Despite the high popularity of mentioned methods, there is a need for new methods development that would allow quick materials fabrication or the synthesis of new, difficult-to-synthesize materials. This work presents two alternative methods of materials fabrication: (1) arc-melting synthesis followed by SPS densification and (2) SHS in SPS synthesis followed by SPS densification. The second part of the presented work focuses on researching a method dedicated to a faster and less expensive synthesis technique (compared to direct synthesis) called self-propagating high-temperature synthesis (SHS). We have already used this method before for magnesium silicide synthesis [17]. Although synthesis of $Bi_{2}Te_{3}$-based materials was discussed in works [18,19] the potential use of the SPS device gives researchers a ready-to-use tool, providing heating and protective atmosphere functionalities. Using the SHS method can reduce the synthesis time from about 30 to about 6 h and does not require additional processing, such as sealing reagents inside a vacuum quartz tube.

Arc melting (AM) was widely used in the metallurgical industry from 1970 [20] but, due to the increased availability of laboratory-size units, this method also gained popularity in materials science over the last few years [21], with success in the fabrication of high-entropy alloys [22,23,24]. The first presented example of the use of SPS is the processing of arc-melted semiconductor half-Heusler material $Hf_{0.6}Zr_{0.4}NiSn_{1-x}Sb_{x}$ and the hafnium-doped cobalt triantimonide semiconductor $Hf_{x}Co_{4}Sb_{11.5}Te_{0.5}$. The second example is the SPS use with SHS, which allows for the quick synthesis of a semiconductor material in just a one-step exothermic reaction. The main advantages of the SHS method are processing time reduction and the possibility of process scale-up. The SHS synthesis method is widely used for SiC synthesis [25,26]. The growing popularity of the SHS method is connected with its multiple adaptations, for example in the synthesis of biomaterials [27] and semiconductors [28].

## 2. Materials and Methods

### 2.1. Materials

The following materials were fabricated or processed using the SPS technique. Additional details can be found in Table 1.
Arc-melted semiconductor materials, *HCST-x* ($Hf_{x}Co_{4}Sb_{11.5}Te_{0.5}$) for *x* = [0, 0.01, 0.02, 0.05, 0.10] and *HZNSS-x* ($Hf_{0.6}Zr_{0.4}NiSn_{1-x}Sb_{x}$ (*half-Heusler*) for *x* = [0.01, 0.02, 0.05])Bismuth telluride-based materials synthesized by SHS technique materials (PBSTS-xxs). The reference material (PBSTq), fabricated inside a quartz vacuum tube, is also included for comparison.

**Table 1 materials-17-01422-t001:** List of fabricated materials.

Identification	Chemical Formula	Processing Method
HCST-01	$Hf_{0.01}Co_{4}Sb_{11.5}Te_{0.5}$	AM, SPS
HCST-02	$Hf_{0.02}Co_{4}Sb_{11.5}Te_{0.5}$	AM, SPS
HCST-05	$Hf_{0.05}Co_{4}Sb_{11.5}Te_{0.5}$	AM, SPS
HCST-10	$Hf_{0.10}Co_{4}Sb_{11.5}Te_{0.5}$	AM, SPS
HZNSS-01	$Hf_{0.6}Zr_{0.4}NiSn_{0.99}Sb_{0.01}$	AM, SPS
HZNSS-02	$Hf_{0.6}Zr_{0.4}NiSn_{0.98}Sb_{0.02}$	AM, SPS
HZNSS-05	$Hf_{0.6}Zr_{0.4}NiSn_{0.95}Sb_{0.05}$	AM, SPS
PBSTq (pREF)	$Bi_{0.5}Sb_{1.5}Te_{3}$	Melting, SPS
PBSTS-01s	$Bi_{0.5}Sb_{1.5}Te_{2.9}Se_{0.1}$	SHS, SPS
PBSTS-06s	$Bi_{0.5}Sb_{1.5}Te_{2.4}Se_{0.6}$	SHS, SPS
PBSTS-12s	$Bi_{0.5}Sb_{1.5}Te_{1.8}Se_{1.2}$	SHS, SPS
PBSTS-18s	$Bi_{0.5}Sb_{1.5}Te_{1.2}Se_{1.8}$	SHS, SPS

As the initial materials, the following pure elements were used: Sb, Bi, Te, Ni (Alfa Aesar, Ward Hill, MA, USA) and Sn (GoodFellow, Huntingdon, UK) with purities better than 99.99%, Se and Co with purity 99.5% (Alfa Aesar, Ward Hill, MA, USA), Hf-(3%Zr) with purity 99.7% (ABCR, Karlsruhe, Germany) and Zr 99.8% (Pol-Aura, Warsaw, Poland).

### 2.2. SPS System

The device used in this work was constructed by the researchers of Łukasiewicz Research Network, Institute of Microelectronics and Photonics and is still maintained and developed by its creators. The operational parameters of the presented equipment are listed in Table 2.

### 2.3. Arc Melting of HCST Cobalt Triantimonide and HZNSS Half-Heusler

The arc-melting method was used to fabricate the alloys containing the elements with high differences in melting points, e.g., hafnium (melting point at 2227 °C) and cobalt (1495 °C). The substrates (in the form of a few pieces of substrate chips) were placed inside a copper reactor and melted in a protective *Ar* atmosphere. During the melting process, the heat energy was delivered to the substrates by an electrical arc. The melting process was repeated 3 to 5 times to ensure chemical homogeneity. The main process parameters were heating current, melting time and cycle count. In this research, a TIG welder with a current range up to 200 A was used as a power source. The obtained materials were further grounded (mortared and ball-milled) and SPS-processed to a form 10 mm diameter, 12 mm height samples. The following parameters were followed:
Cobalt triantimonide: T = 650 °C, P = 50 MPa, time = 15 min.Half-Heusler: T = 1000 °C, P = 50 MPa, time = 25 min.

The samples were sanded (to remove graphite from the surface), polished and cut for further materials characterization.

### 2.4. Process Data Collection and Device Upgrade

The SPS system upgrade model, presented in the Figure 3, is aimed to extend the control of process parameters while monitoring the values over time. Figure 4 shows the collected parameters before (a) and after (b) the device upgrade. The following changes were introduced:
Acquisition of power supply parameters (voltage and current), allowed us to gain information about delivered power and, in some cases, the resistance of the sintering system.Constant stabilization of the pressing force. A hysteresis regulator was used previously, causing lower reproducibility of the pressure value over time. Now, a 5-way proportional valve driven by a quasi-logic controller is installed, resulting in high stability and reproducibility of the pressure.Up to 6 thermocouples (labeled as #1T1, #1T2, #2T1, #2T2, #3T1, #3T2) can be connected to determine the temperature gradient across the sample up to 950 °C. Figure 3 presents an example of thermocouple placement, where #1T1 is a process control thermocouple, and other sensors collect temperatures from different parts of the system allowing us to determine the temperature gradient. Before the upgrade only one temperature was acquired.Above 500 °C, infrared temperature measurement can be used. The infrared curve shows when the temperature exceeds 500 °C (Figure 5).

**Figure 3 materials-17-01422-f003:**
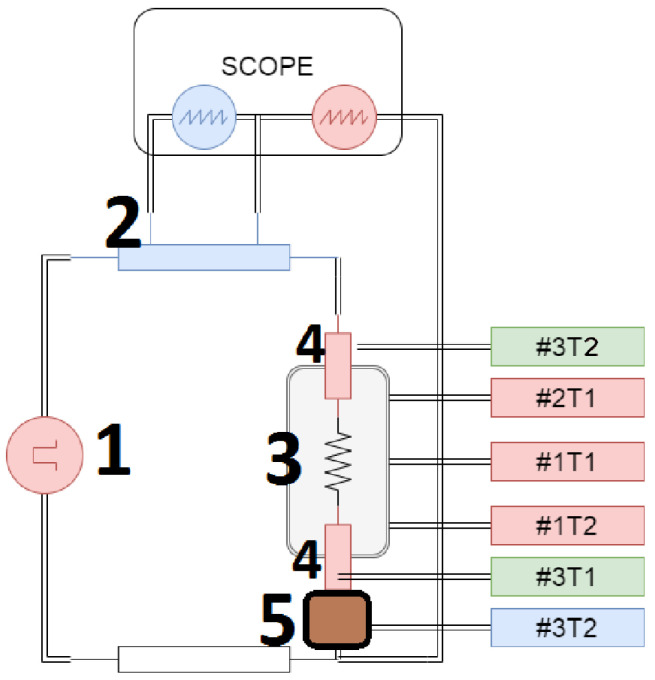
Schematic diagram of modernized SPS device: 1—alternate current power supply, 2—current sensing resistor, 3—graphite die with sample, 4—upper/lower punch, 5—graphite spacer.

**Figure 4 materials-17-01422-f004:**
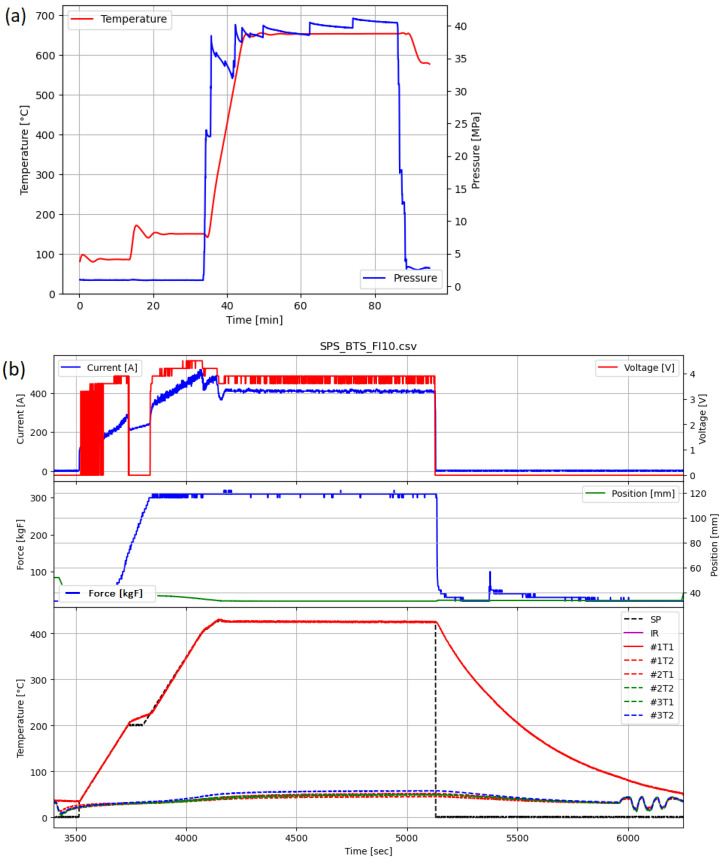
Data collected (**a**) before and (**b**) after apparatus development.

Extending the capabilities (both process control and process parameter monitoring) enabled conducting nonstandard processes, such as the SHS reaction, and allowed us to gain detailed process information.

### 2.5. SHS

SHS was conducted inside a graphite reactor placed in the SPS apparatus. The stoichiometric amounts of elements were homogenized using an automated agate mortar and heated up to 500 °C in an inert gas (Ar) atmosphere. Figure 5 shows the detailed heating program of the SHS reaction. There is a visible voltage drop at Time = 250 s, which is implied by the exothermic reaction. The process consists of two stages realized in one process:
Heating the system to 375 °C and maintaining this temperature for 60 s. This step realizes the first, exothermic stage of materials synthesis (SHS occurs during the first initial heating).Heating the system to 475 °C and maintaining this temperature for 300 s. This step was applied to fully react reagents and homogenize material.

**Figure 5 materials-17-01422-f005:**
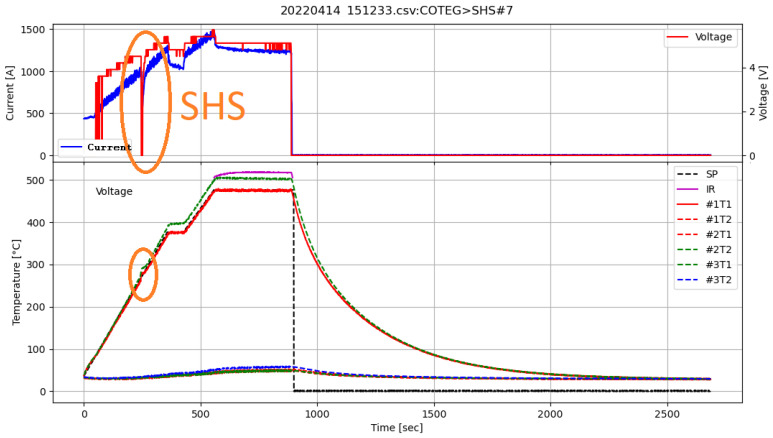
Full temperature and power supply parameters during the SHS process in SPS apparatus. The chart includes the Voltage, Current—power supply readings, SP—Temperature Set Point, IR—Pyrometer temperature, and #xTy—thermocouple reads.

The process progress analysis allowed the determination of an exothermic reaction occurrence by observing a significant drop in power delivered by the device. After synthesis, the samples were grounded and sintered (450 °C, 50 MPa, 15 min) to 10 mm diameter and 12 mm height samples and characterized.

### 2.6. XRD Diffraction

Phase composition studies using the X-ray diffraction technique were carried out using a Brucker D8 Advance device equipped with a Cu anode. Measurements were performed in the range from 10° to 120° (0.025° step), with a counting time of 3 s. XRD diffraction was performed on the 1 mm height 10 mm diameter samples previously inspected for thermal conductivity (PBSTS01ss—**S**HS—**s**olid). Analysis of materials between stages of fabrication (PBSTS01sr, PBSTS01sp) were performed on powders after homogenization (sr—**S**HS—**r**aw material) and SHS (sp—**S**HS—**powder**).

### 2.7. Thermal and Electrical Parameter Characterization

Thermal conductivity was measured using the LFA 457 apparatus (Netzch, Selb) [29]. The measurement was performed on a 10 mm diameter 1 mm height disc in the inert Ar protective atmosphere in the temperature range RT–700K. The surfacse of samples were polished and a graphite layer was sprayed to ensure the emmisity factorn was constant for all samples.

The electrical properties measurements were performed using the SeebTest device, PESS, Kraków, Poland using the 4-wire method, according to the concept presented in Figure 6.

The measurements were performed on cylindrical samples with 10 mm diameters and heights ranging from 10 to 14 mm. Electrodes $T_{h}$ and $T_{c}$, placed in previously drilled holes with a precisely defined distance $L_{x}$, allowed for simultaneous measurement of the voltage drop on the sample. The measurements were performed within a defined temperature range. The heating elements $H_{h},H_{c}$ stabilized the sample temperature. If the temperature of sample was stable, the voltage drop on the sample was measured as a result of the alternating change in the direction of the predefined current flow. The heating elements then induced a temperature gradient ΔT to determine the Seebeck coefficient α. The measurement was performed with a step of 25 °C during heating and cooling.

## 3. Results

### 3.1. Thermoelectric Properties of Arc-Melted and SPS-Processed Materials

Figure 7 presents the electric properties of hafnium-doped $Co_{4}Sb_{11.5}Te_{0.5}$. The absolute value of Seebeck coefficient increased in samples HCST-05 and HCST-10. The addition of lower amounts of hafnium had no visible impact on the Seebeck coefficient. An increase in hafnium content caused slight decrease in electrical conductivity.

For the materials from the $Hf_{x}Co_{4}Sb_{11.5}Te_{0.5}$ series obtained with arc melting, the doping effect is visible for x=0.05 and x=0.10. With increasing Hf content, we observe a slight increase in the Seebeck coefficient and a decrease in electrical conductivity (Figure 7). However, the noticeable reduction in thermal conductivity, presented in Figure 8, occurs only for x=0.10. Figure 8 shows that in the cases of HCST-05 and HCST-10 materials, they were characterized with slightly higher ZT parameters.

Figure 9 presents the electrical parameters of HZNSS half-Heusler materials. A slight Seebeck coefficient and electrical conductivity drop can be observed with the addition of Sb.

Figure 10 presents the thermal conductivity and ZT parameters of HZNSS samples. The thermal conductivity of HZNSS samples dropped with the increase in Sb content (Figure 9a). The ZT parameter of HZNSS materials exhibited similar behavior below 500 K, but above this temperature the sample with high Sb content exhibited significantly lower ZT values (Figure 9b).

### 3.2. SHS Results

The following section is subdivided into two paragraphs, describing the process of SHS and presenting fabricated materials properties.

#### 3.2.1. Analysis of the SHS Stages

The effectiveness of the synthesis was confirmed based on the analysis of the X-ray diffraction of the material; Figure 11 shows the diffractogram of the material at various stages of synthesis.

The analysis of the diffractogram of the powder after SHS showed partial reaction of the substrates—both diffraction lines from the substrates and the product are visible. After regrounding and sintering, the material exhibited a high degree of conversion; XRD analysis did not show the presence of significant amounts of starting elements. The sample was annealed at 200 °C for 2 h to test the temperature stability of the material—the last diffractogram showed no changes in the phase composition of the compound.
PBSTS01sr—The material after the homogenization process in an automatic mortar. This is the analysis of the phase composition, in addition to confirming the presence of pure elements, i.e., tellurium (00-004-0555), antimony (00-077-3384) and bismuth (01-078-6571).PBSTS01sp—The fragmented material after synthesis *SHS* consisted mainly of the phase $\left(Bi_{0.58}Sb_{1.42}\right)\left(Se_{0.12}Te_{2.88}\right)$ (01-082-7905) and bismuth oxide (01-078-0736). Some of the reflections are difficult to distinguish due to their overlap.PBSTS01ss—The material after the *SPS* technique contained the $Bi_{0.5}\left(Sb_{1.5}Te_{3}\right)$ phase and probably bismuth oxide (01-078-0736).

Figure 12 presents XRD patterns for all four PBSTS samples after SPS. The PBSTS01ss sample contained mostly the $Bi_{0.5}Sb_{1.5}Te_{3}$ phase, small amounts of $\left(Bi_{0.58}Sb_{1.42}\right)\left(Se_{0.12}Te_{2.88}\right)$ and oxide phases. The increase in Se content caused the appearance of multiple Bi-Sb-Te-Se phases and an oxide phase.

#### 3.2.2. Thermoelectric Properties of SHS-Processed Bismuth Telluride

The results of Seebeck coefficient measurements for the PBSTq material (Figure 13a) showed high stability of the Seebeck coefficient as a function of temperature, in the range of 50–150 °C. The introduction of a small amount of selenium into the structure caused an increase in the Seebeck coefficient by 10% (for the PBSTS01s composition in relation to the undoped PBSTq material), but a further increase in the selenium content resulted in a significant decrease in the Seebeck coefficient. All materials synthesized using the SHS method exhibited 3 times lower electrical conductivity than the reference sample (Figure 13b).

The PBSTS samples thermal conductivity decreased with increasing the selenium content with the only exception of sample PBSTS01s. The ZT factor of PBSTS01s sample equals about half the value of the reference material (Figure 14).

## 4. Discussion

The modernized SPS device allowed for gaining information about process dynamics, which was key information during the optimization of the SHS parameters. Most SPS devices track only three basic parameters: maximum temperature, time and applied pressure [2,30]. An increased number of tracked parameters allowed for continued and detailed process monitoring, which was crucial for understanding the phenomena and stages of the sintering.

### 4.1. The Properties of Arc-Melted Materials

Only the highest Hf addition resulted in a visible thermal conductivity drop (Figure 8) —this suggests that the hafnium influences mostly micro-structure of the material, with a smaller affect on the electron structure. The HZNSS materials exhibited similar thermoelectric properties, although the Seebeck coefficient, electrical and thermal conductivity dropped slightly with an increase in Sb content (Figure 9).

Due to low concentrations of hafnium (HCST) and antimony (HZNSS), it was nearly impossible to examine phase differences within fabricated materials using XRD or SEM methods, but due to Seebeck coefficient sensitivity on chemical and phase composition it was possible to distinguish differences between the HCST-[01,02] and HCST-[05-10] materials.

### 4.2. SHS

Conducted SHS allowed for the fabrication of a p-type bismuth telluride material. Despite how oxide phases were identified in all materials fabricated using the SHS method, the analysis of the ZT factor (Figure 14) shows that it is possible to reach the ZT≃0.5 with a significant reduction in processing time and energy consumed during synthesis.

### 4.3. Thermoelectric Properties of SHS Materials

Compared to the results presented in the work [31], the values of the Seebeck coefficients obtained in the presented work for *p*-type materials are less dependent on the temperature [31]. A noteworthy achievement is the PBSTq material, which is characterized with an almost constant, stable over time and temperature α coefficient for the full temperature range.

The p-type material obtained as part of the work, undoped with selenium (pREF), had a Seebeck coefficient ranging from 165 $\frac{\mu V}{K}$ to 185 $\frac{\mu V}{K}$, which is approximately 20$\frac{\mu V}{K}$ higher than those presented in the work [32]. The nature of the Seebeck coefficient dependence is also similar for the *Bi*-*Sb*-*Te*-*Se* compound with a small amount of selenium, although in this case an increase in the coefficient value of approximately 40 $\frac{\mu V}{K}$ was observed (Figure 13a). The similarity of properties during heating and cooling ensures material temperature–time stability.

In the work [33], the material $Bi_{0.5}Sb_{1.5}Te_{3-x}Se_{x}$ was tested in the range 0↔ 0.1. Modification of the material with selenium in the tested range resulted in the standardization of the Seebeck coefficient in the target operating temperature range (the characteristics of the Seebeck coefficient are shown in work [33]). The authors of [33] indicate the possibility of increasing the Seebeck coefficient by introducing a 3% surplus of tellurium. The amount of selenium in the range 0.01<x<0.1 does not affect the thermal properties of the material, increasing the Seebeck coefficient by 10%.

Further increase of the Se addition in this work resulted in a decrease in the Seebeck coefficient of the material, which was caused by the formation of undesirable *n*-type semiconductor areas in the *p*-type material. The selenium addition above x>0.1 decreases the thermoelectric properties of material, lowering Seebeck coefficient and electrical conductivity. This results in a ZT factor drop below 0.1, making the materials with higher selenium content not suitable for use for electricity generation. Although the research shows that it is possible to synthesize bismuth telluride-based materials using an SPS device, more studies are required (including lowering selenium content and verifying other dopants).

## 5. Conclusions

This publication shows the versatility of SPS devices, presenting examples of use, extension and adaptation of the SPS system for synthesis and fabrication of semiconductor materials. The multi-point measurement allowed for precise control of process parameters and provided much additional information about the process.

## Figures and Tables

**Figure 1 materials-17-01422-f001:**
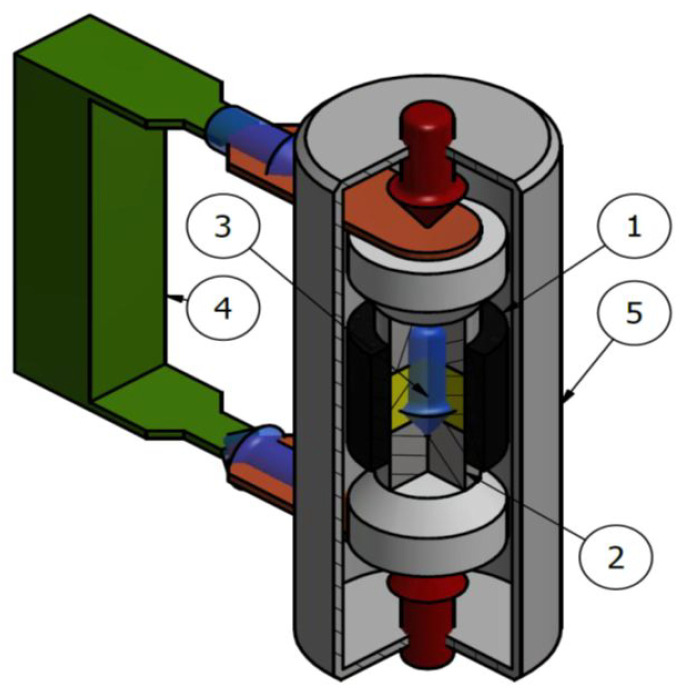
Scheme of the system for spark sintering using the SPS technique: 1—die, 2—punch, 3—sintered material, 4—power supply, 5—vacuum chamber.

**Figure 2 materials-17-01422-f002:**
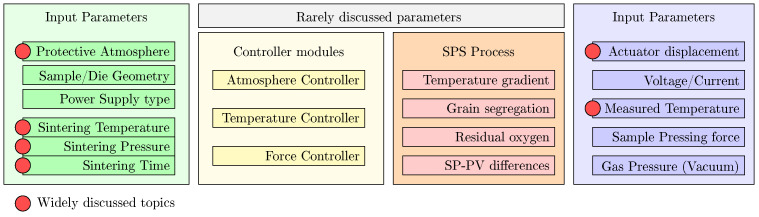
Parameters of SPS processes described in [2].

**Figure 6 materials-17-01422-f006:**
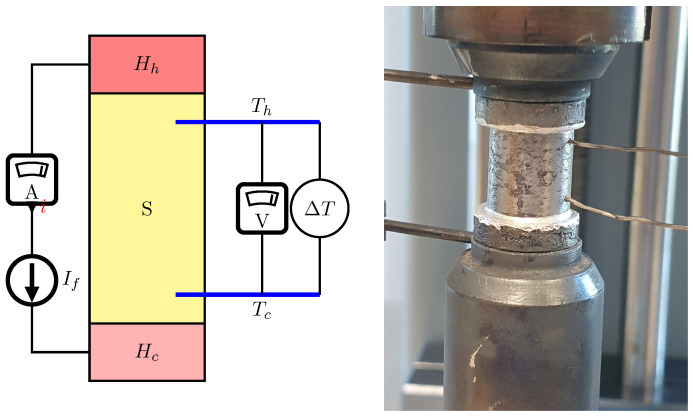
Scheme of the method implemented in the SeebTest device and a photo of the holder with the sample in the device, *S*—sample, $H_{h}$, $H_{c}$—upper/lower heater.

**Figure 7 materials-17-01422-f007:**
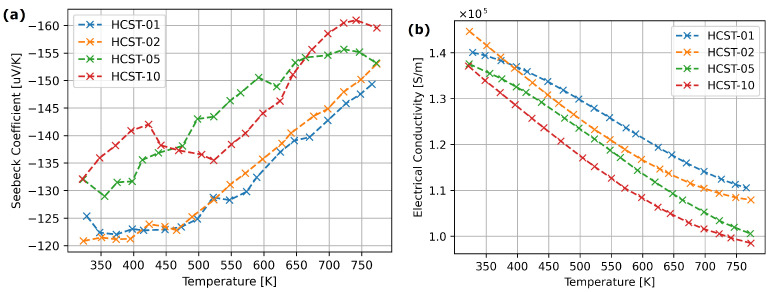
Transport properties of hafnium-doped cobalt triantimonide: (**a**) Seebeck coefficient, (**b**) electrical conductivity.

**Figure 8 materials-17-01422-f008:**
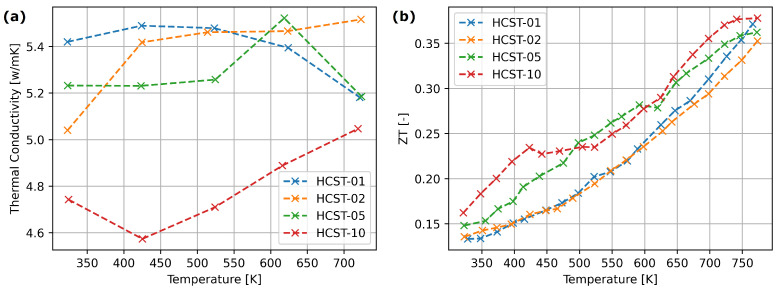
(**a**) Thermal conductivity. (**b**) Figure of Merit (ZT) factor of synthesized HCST thermoelectric materials.

**Figure 9 materials-17-01422-f009:**
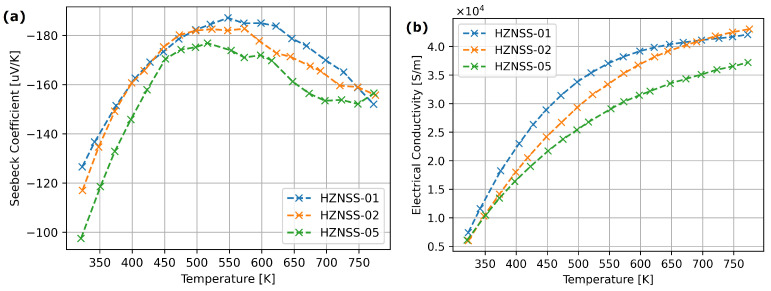
Transport properties of HZNSS half-Heusler materials: (**a**) Seebeck coefficient, (**b**) electrical conductivity.

**Figure 10 materials-17-01422-f010:**
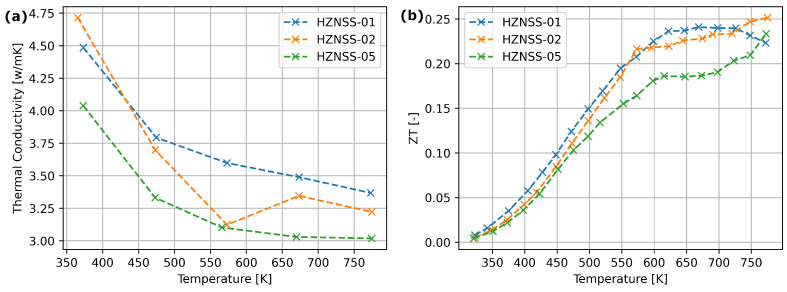
(**a**) Thermal conductivity. (**b**) Figure of Merit (ZT) factor of synthesized thermoelectric materials.

**Figure 11 materials-17-01422-f011:**
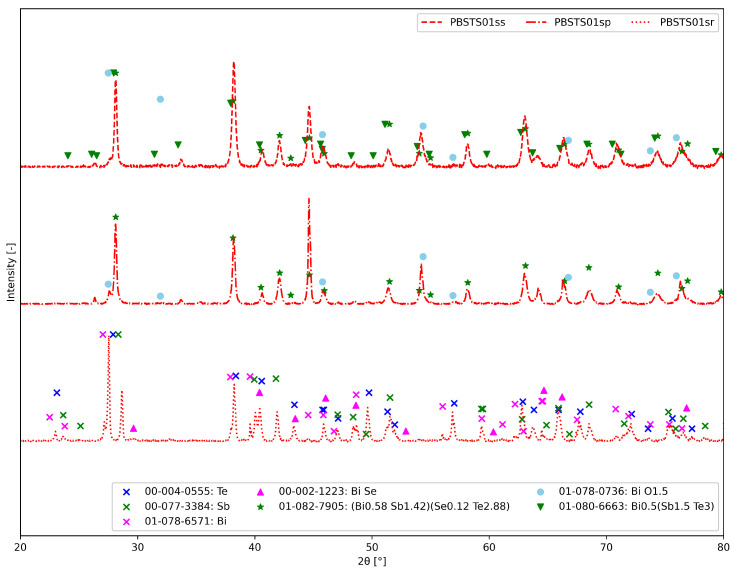
XRD patterns of material at different stages of synthesis: PBSTS01sr—powder before SHS, PBSTS01sp—powder after SHS; PBSTS01ss—solid material after SPS.

**Figure 12 materials-17-01422-f012:**
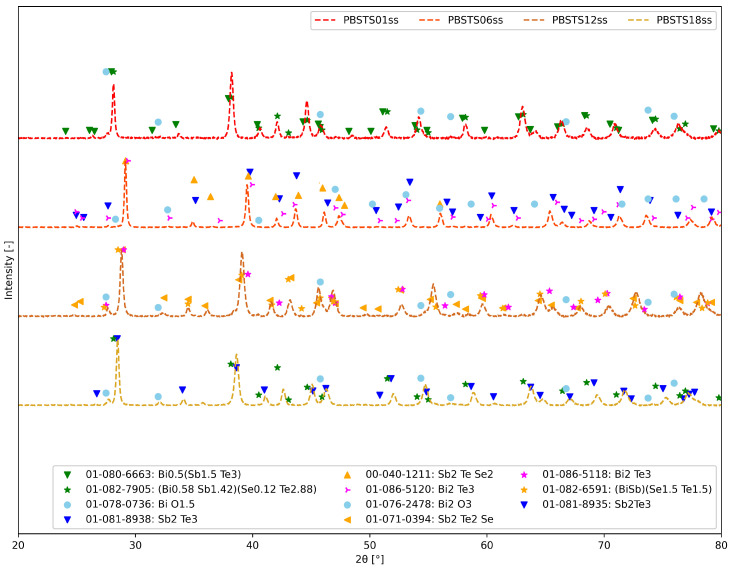
XRD patterns of samples with different selenium contents.

**Figure 13 materials-17-01422-f013:**
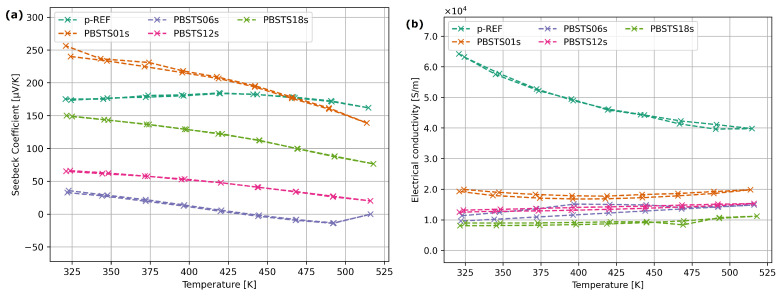
Transport properties of SHS-processed PBSTS materials: (**a**) Seebeck coefficient, (**b**) electrical conductivity of SHS-processed bismuth telluride samples with different selenium contents.

**Figure 14 materials-17-01422-f014:**
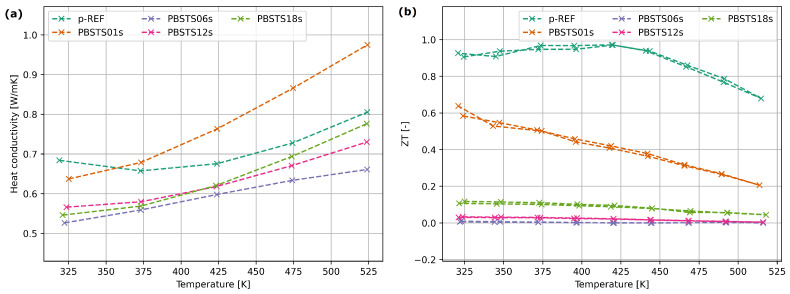
(**a**) Thermal conductivity. (**b**) Figure of Merit of SHS bismuth telluride samples with different selenium quotients.

**Table 2 materials-17-01422-t002:** Presented apparatus operational parameters.

Parameter	Value/Range	Unit
Operating temperature	RT-2000	°C
Sample diameter	10–50 ^1^	mm
Power supply max. current	5000	A
Power supply max. voltage	10	V
Power supply type	Alternate current	
Ultimate vacuum	$10^{-5}$	mBar
Pressing force (max)	10	Tons

^1^ Samples with higher diameters can be also processed.

## Data Availability

Data are contained within the article.

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
