# Peer review of "Using the Spark Plasma Sintering System for Fabrication of Advanced Semiconductor Materials"

_materials, 2024, doi:10.3390/ma17061422_

Round 1

Reviewer 1 Report

Comments and Suggestions for Authors

The paper presents results about fabrication of materials by SPS. The process parameters and characterization are well presented and commented. I suggest only small revisions about the description of the process (section Materials and methods). 

The authors use the words synthesis, process and sintering to describe their protocol. The word synthesis generally refers to the step of powder synthesis and sintering is the densification of powder compact. Should the authors clarify the use of these words to describe the different steps of materials' processing ?

In the Materials and methods section, it should be more appropriate to present first the different fabrication protocols (i.e. arc-melting and SHS) and then the characterizations. Then, the authors could detail which type of samples was characterized by XRD (powders or densified samples) ?

In the subsection 2.1, the authors could detail the raw materials used for the powder synthesis. It could help to understand the evolution of the phases visible in the XRD patterns. 

The subsection 2.6 presents upgrade of the SPS device. It is not clear for me if the processing presented in subsection 2.1 to 2.6 was made with the upgraded version of the SPS apparatus. Can the authors clarify this point ? 

Can the authors correct the sentence: The samples HCST-05 and HCST-10 showed and increasement of absolute values of Seebeck coefficient ?

Author Response

Thank You for constructive comment.

Please find our reply attached.

Reviewer 2 Report

Comments and Suggestions for Authors

Dear authors,

in your article you present a self-built SPS device and you compare this technique to other compacting techniques (Arc melting and SHS) by using material examples like HCST, HZNZZ and PBSTS.

For publishing in this journal the quality of the manuscript has to be significantly improved. For the reader it is very difficult to understand what the goal of this study was. Did you aim for improving the quality of the materials? Do you want to present an upgraded version of SPS device? Do you want to compare different compaction techniques? Focus on one of these questions and simplify the manuscript.

Some detailed comments below:

1) Check the title: SPS is the abbreviation for Spark plasma sintering. Remove the additional "sintering"

2) Abstract: Introduce the abbreviation for SHS

3) Formula 1: Mention that heat conductivity consists of two parts: electronic and lattice. This makes it difficult to improve the efficiency of Thermoelectrics (=increase zT value).

4) Line 144: Cu lamp should be Cu anode or Cu radiation

5) For XRD/LFA measurements: Are you polishing the samples before to ensure you have a carbon free surface (no contamination from the graphite die?) Which side of the pellet is used for the measurement? Always the same side? Did you detect differences between upside and downside? (Because it is well known that due to the electric polarity during SPS ions can move (separation) leading to different stoichiometry and composition)

6) Related to 5): What is the carbon content in general? Did you check the chemical composition after compaction using EDS or XPS? I can only see nominal composition.

7) What about reproducibility? How many pellets did you produce? How often did you measure? Did you measure different positions (especially for LFA but also for XRD?)

8) Figure 8 is not discussed in the text.

9) Improve the quality of all graphs and figures. Especially the XRD patterns figure 11 and 12. The location of phases overlap with the graph and it is really hard to understand this analysis. Additionally: You claim that you see the BiSe phase, which I don't (?)

Comments on the Quality of English Language

Dear editiors,

the authors present a self-built SPS device and compare this technique to other compacting techniques (Arc melting and SHS) by using material examples like HCST, HZNZZ and PBSTS. The study is very complicated to follow because of the unclear message that the authors want to communicate. Additionally, I found many issues regarding the experimental design and the presentation of the results. For more information please find my detailed answer to the authors.

Therfore, I don't recommend publishing this article in the current state. It needs significant improvements and can be reconsidered for publication after major revision.

Author Response

Thank You for Your constructive comments and advices. The article was revised due to suggestions. Please find our comments attached.

Round 2

Reviewer 2 Report

Comments and Suggestions for Authors

Dear authors, thank you for revising the mansucript and answering my comments. The quality of the manuscript improved a lot and can be considered for publication.

Comments on the Quality of English Language

minor editing necessary